# Autoreactivity of Broadly Neutralizing Influenza Human Antibodies to Human Tissues and Human Proteins

**DOI:** 10.3390/v12101140

**Published:** 2020-10-08

**Authors:** Surender Khurana, Megan Hahn, Laura Klenow, Hana Golding

**Affiliations:** Division of Viral Products, Center for Biologics Evaluation and Research (CBER), FDA, Silver Spring, MD 20993, USA; megankhahn@gmail.com (M.H.); laura.klenow@fda.hhs.gov (L.K.); hana.golding@fda.hhs.gov (H.G.)

**Keywords:** influenza, hemagglutinin, autoreactivity, polyreactivity, stem, head, vaccine, antibody affinity, bNAbs, universal influenza

## Abstract

Broadly neutralizing monoclonal antibodies (bNAbs) against conserved domains in the influenza hemagglutinin are in clinical trials. Several next generation influenza vaccines designed to elicit such bNAbs are also in clinical development. One of the common features of the isolated bNAbs is the use of restricted IgV_H_ repertoire. More than 80% of stem-targeting bNAbs express IgV_H_1-69, which may indicate genetic constraints on the evolution of such antibodies. In the current study, we evaluated a panel of influenza virus bNAbs in comparison with HIV-1 MAb 4E10 and anti-RSV MAb Palivizumab (approved for human use) for autoreactivity using 30 normal human tissues microarray and human protein (>9000) arrays. We found that several human bNAbs (CR6261, CR9114, and F2603) reacted with human tissues, especially with pituitary gland tissue. Importantly, protein array analysis identified high-affinity interaction of CR6261 with the autoantigen “Enhancer of mRNA decapping 3 homolog” (EDC3), which was not previously described. Moreover, EDC3 competed with hemagglutinin for binding to bNAb CR6261. These autoreactivity findings underscores the need for careful evaluation of such bNAbs for therapeutics and stem-based vaccines against influenza virus.

## 1. Introduction

Concerted efforts are underway to develop broadly protective therapeutic antibodies and cross-reactive vaccines against influenza. The development of such next-generation (or universal) vaccines was greatly energized by the isolation of broadly neutralizing human monoclonal antibodies (bNAbs) that target relatively conserved epitopes in the hemagglutinin (HA). Such bNAbs and vaccines designed to generate such antibodies could provide protection against drifting strains over longer periods compared with the current seasonal vaccines, as well as against avian influenza strains with pandemic potential [1,2,3]. The conserved targets of the isolated bNAbs mapped either to the stem region or to the globular head of the HA including the receptor-binding site (RBS) [4,5,6].

Interestingly, many of the bNAbs targeting the conserved stem region of influenza virus hemagglutinin show restricted usage of IgH V(D)J sequences that are derived from the *V_H_1-69* family. In several human anti-stalk antibodies recognizing Gp1 and Gp2 influenza A viruses, the *V_H_1-69* was heavily mutated as a consequence of somatic hyper mutation (SHM), which conferred high affinity binding to the overlapping membrane-proximal stalk domain. However, IgV_H_1-69 gene segment is also associated with polyreactive responses in autoimmune pathologies and with certain B-cell cancers. Interestingly, several HIV-1-specific bNAbs demonstrated propensity to be polyreactive and/or autoreactive.

In the case of influenza antibodies, previous studies described polyreactivity of MAbs to some proteins in the absence or presence of BSA [7,8], but the methods used in these studies do not mimic physiological conditions in vivo. It is critical to explore autoreactivity of MAbs to human tissues and human proteins in the presence of human serum, which is the natural milieu in vivo. Moreover, earlier studies did not look at the impact of binding of the human proteins on the interaction of the bNAbs with its cognate influenza virus hemagglutinin. Therefore, we evaluated the autoreactivity of a panel of influenza virus bNAbs in comparison with the anti-RSV antibody palivizumab, which is approved for prophylactic treatment of infants that does not show autoreactivity [9]. Analysis of human tissue microarrays (30 normal tissues derived from each of 3 donors) microarray and of protein microarrays containing over 9000 human proteins revealed several bNAbs that reacted with human tissues and human proteins, while only MAb CR6261 [7,8] bound with high affinity to an autoantigen “Enhancer of mRNA decapping 3 homolog” (EDC3) [10]. This autoantigen was also identified by a similar screen reported by Bajic et al. [8]. However, in the current study we demonstrate that EDC3 binding of CR6261 blocked antibody binding to its cognate influenza hemagglutinin in surface plasmon resonance (SPR) competition assay. The potential of auto-reactivity due to molecular mimicry or other mechanisms, should be further evaluated. Requires careful evaluation of such bNAbs and vaccines intended to generate such bNAbs.

## 2. Materials and Methods

### 2.1. Tissue Microarray

Tissue microarrays with 30 different human normal tissue types and 3 donors per tissue type of adrenal gland, bone marrow, breast, cerebrum, pituitary gland, colon, heart, kidney, liver, pancreas, placenta, prostate, salivary gland, small intestine, cerebellum, esophagus, lung, mesothelial cell, ovary, peripheral nerve, skin, spleen, skeletal muscle, stomach, testis, thymus, thyroid, tonsil, uterus, and cervix were obtained from BioChain. This standard tissue array with 90 tissue samples designed in conformance with FDA guidelines and meeting the requirements for IHC (immunohistochemistry) and IVD (in vitro diagnostic devices) certification was stained with antibodies, and individual tissue on the slides were used for examination.

### 2.2. Semiquantitative Score of IHC

All immunohistochemistry stained slides were digitally scanned by Nanozoomer XR slide-scanning system (Hamamatsu Photonics K.K., Shizuoka, Japan) and stored as ndpi files for further analysis. Each tissue microarray specimen was blindly scored based on the reactivity from negative (=0), mild (=1), moderate (=2), or strong (=3) positive. The scores from 3 tissues samples were averaged to obtain the mean score for antibody reactivity to the individual tissue. The scored data were analyzed by Microsoft excel and Prism 7 (GraphPad software, La Jolla, CA, USA).

### 2.3. Production of Recombinant Human MAbs

For IgG production, the genes for the heavy- and light-chain (kappa or lambda) variable domains were synthesized and cloned into Abvec-hlgG1, AbVec-hIgKappa, or AbVec-hIgLambda protein-expression vectors as appropriate containing human heavy- and light-chain constant domains. These expression plasmids were a kind gift by Dr. Patrick Wilson (University of Chicago). Briefly, the bNAb V(D)J sequences were cloned into vectors containing human IgG1 framework using conventional endonuclease restriction digestion, as previously reported [11]. IgGs were produced by transient transfection of suspension HEK293F cells using polyethylenimine (PEI; Polysciences, Warrington, PA, USA). Supernatants were harvested 5 days later, clarified by centrifugation, and IgGs were purified using Protein A Plus Agarose (Thermo Fisher Scientific, Waltham, MA, USA) as per manufacturer’s instructions. Each MAb was qualified by testing for binding to influenza virus HA by SPR.

### 2.4. Autoreactivity Protein Binding by ProtoArray Analysis

Recombinant human IgG1 MAbs were screened for binding on protein microarrays (ProtoArray: #PAH0525101; Invitrogen, Carlsbad, CA, USA) precoated with >9000 human proteins in duplicate from a single lot of arrays. Human MAbs were diluted to 10 µg/mL final concentration in human immunoglobulin G-depleted serum (BEI; NR-49447). The binding patterns of human anti-HA bNAbs were compared with the human myeloma protein 151 K diluted to 10 µg/mL final concentration in human immunoglobulin G-depleted serum (BEI; NR-49447). Array-bound anti-human IgG served as the loading control for the detection Ab, as well as for the secondary reagent. The ProtoArray microarray (Invitrogen, Carlsbad, CA, USA) was blocked and incubated with 10 μg/mL of MAb or isotype control 151 K. MAb binding to array protein was detected with 1 μg/mL of Alexa Fluor 647-labeled anti-human IgG secondary antibody (Invitrogen). The ProtoArray microarrays were scanned using a GenePix 4000B scanner (Molecular Devices, San Jose, CA, USA) at 635 nm. Z scores were calculated as the number of standard deviations of the signal from the mean of the corresponding MAbs. After scoring, stringent quality assessment was undertaken, including high correlation coefficients of duplicate spots of printed proteins (average *r* = 0.92). MAbs were screened for reactive antigens on protein microarrays following the manufacturer’s instructions and selected with Z-Score > 3, Z-Factor > 0.5, Signal ratio to 151 K > 5, CI *p*-value < 0.05, and CV < 0.5.

### 2.5. SPR Based HA Binding and Competition Assay

Steady-state equilibrium binding of antibody with different proteins was monitored at 25 °C using a ProteOn surface plasmon resonance (BioRad, Hercules, CA, USA). The purified MAbs were captured on Protein G sensor chip with 200 resonance units (RU) in the test flow channels (BioRad, Hercules, CA, USA). Binding activity of all MAbs was analyzed by binding to the purified recombinant HA protein from either H1 (A/California/04/2009; NR15749, or BEI) or H3 (A/Brisbane/10/2007; NR-19238; BEI) strains or the selected protoarray proteins (APEH (Abnova, Taipei, Taiwan), EDC3 (Origene, Rockville, MD, USA), MAPK6 (Origene, Rockville, MD, USA), STYX (Origene, Rockville, MD, USA), TRIM21 (Origene, Rockville, MD, USA), STAT4 (Sino Biologicals, Chesterbrook, USA), JO-1 (Origene, Rockville, MD, USA), and Thyroglobulin (Abcam, Cambridge, UK). Samples of 500 μL of freshly prepared proteins at 20 μg/mL and 2 μg/mL in BSA-PBST buffer (PBS pH 7.4 buffer with Tween-20 and BSA) were injected at a flow rate of 50 μL/min (contact duration, 180 s) for association. Responses from the protein surface were corrected for the response from a mock surface and for responses from a buffer-only injection were calculated with BioRad ProteOn manager software (version 3.0.1). The maximum resonance units (Max RU) data shown for each antibody binding in the manuscript figures were calculated by multiplying the observed RU signal with the dilution factor for each protein sample, to provide the binding signal for 20 μg/mL protein concentration. All SPR experiments were performed at least twice and the researchers performing the assay were blinded to sample identity. In these optimized SPR conditions, the variation for each sample in duplicate SPR runs was <5%.

Antibody-affinity constant for the binding of EDC3 with CR6261 were determined directly for serial dilutions (2, 0.4, and 0.08 μg/mL) of EDC3 with CR6261 for sensorgrams with Max RU between 5 and 100 RU using SPR (as described above) and calculated using the BioRad ProteOn manager software. The EDC3-MAb binding SPR experiments were also performed in the presence of human immunoglobulin G-depleted serum- PBST buffer that yielded similar binding kinetics.

For competition experiment, various proteins at 20 µg/mL were injected for 180 s at 50 μL/min on MAb coated sensor chip, followed by injection of recombinant HA (10 μg/mL) for 120 s for association. Responses from the protein surface were corrected for the response from a mock surface and for responses from a buffer-only injection. Total % HA binding remaining in the presence of different host proteins were calculated with Bio-Rad ProteOn Manager software (version 3.0.1).

## 3. Results

### 3.1. Tissue Microarray Immunohistochemistry for Anti-Influenza Virus Hemagglutinin bNAbs

The goal of this study was to determine if influenza specific bNAbs are more likely to be autoreactive to human tissues and to identify the interacting host proteins. To that end, we produced a panel of previously described bNAbs whose targets were mapped either to the stem region (CR8020, CR6261, CR9114, F10, F16, and F2603) [4,5,12] or to the receptor-binding domain (CH65) [6] of the influenza virus hemagglutinin (HA). As internal comparators, we used the anti-HIV-1 bNAb 4E10, which has known autoreactivity, as a positive control) [13] and FDA-approved anti-RSV F human MAb Palivizumab, as negative control [9].

In the first study, a human tissues microarray with 30 different normal human tissues each from 3 different donors was used to determine the autoreactivity of the bNAbs (Appendix A). This standard tissue array with 90 tissue samples designed in conformance with FDA guidelines were stained with antibodies and met the requirements for IHC (immunohistochemistry) and IVD (in vitro diagnostic devices) certification. All tissue microarray specimens were scored blindly, with scores ranging from negative (=0), mild (=1), moderate (=2), or strong (=3) reactivity. The scores from triplicate tissue samples were averaged to obtain the mean score for antibody reactivity to a given tissue. The reactivity of individual MAbs against all the tissues is shown in Table 1.

The heat map correlates with the mean score of reactivity of a given antibody against each tissue. The positive control MAb 4E10 (anti-HIV-1 gp41-MPER) reacted with the largest number of tissues and demonstrated the strongest binding to pituitary gland cells (various staining pattern) and acinar cells of pancreas. MAb 4E10 also showed mild to moderate positive scores with adrenal gland, cardiac myocytes, tubular cells of kidney, epithelium of prostate, and small intestine. In contrast, Palivizumab showed negligible reactivity with all tissues (Table 1). Nonspecific staining was not observed in these experiments.

Among the influenza specific bNAbs, tissue reactivity was either very low/negative (CR8020 and CH65), or low to modest to few tissues for the other bNAbs. The strongest reactivity was observed with bNAbs CR6261, CR9114, and F2603 against pituitary gland tissue (score of 1.33, 1.67, and 1.50, respectively) (Table 1).

### 3.2. Autoreactivity to Human Proteins by ProtoArray Analysis

To identify the human proteins interacting with these bNAbs, all the recombinant human IgG1 MAbs were screened for binding to protein microarrays (ProtoArray; Invitrogen) containing >9000 human proteins in duplicates. To mimic the natural milieu of human antibodies in vivo, all bNAbs were diluted to 10 µg/mL (final concentration) in immunoglobulin G-depleted human serum to reduce nonspecific binding. The binding patterns of human anti-HA bNAbs were compared with the human myeloma protein 151 K, which binds to a known autoantigen, betaine-homocyteine methyltransferase 2 (BHMT2) [13] (representative data is shown in Appendix A). Array-bound anti-human IgG served as the loading control for the detection antibody, as well as for the secondary reagent. MAbs were screened for reactive antigens on protein microarrays following the manufacturer’s instructions and selected with Z-Score > 3, Z-Factor > 0.5, Signal ratio to 151 K > 5, CI *p*-value < 0.05, and CV < 0.5 for high stringency.

As expected, all the human IgG MAbs bound to TRIM21, a known cytosolic Fc receptor [14] (Figure 1A).

In addition, we identified broad low to modest reactivity of some of the bNAbs (compared with 151K, Appendix A) with several autoantigens listed in Appendix A, including APEH, STYX, STAT4, MAPK6, and Jo-1. The reactivity pattern of these ProtoArray-identified proteins was analyzed further and was confirmed using SPR (Figure 1A). Interestingly, under the stringent conditions used, a strong specific binding for bNAb CR6261 to an autoantigen Enhancer of mRNA decapping 3 homolog (EDC3) [10], (CR6261 signal ratio to 151K = 221 in ProtoArray) was observed, which was previously also identified by Bajic et al. [8] (Figure 1A, black bar). The affinity of MAb CR6261 binding to EDC3 was calculated to be 44.5 nM (Figure 1B).

### 3.3. EDC3 Competes with HA for Binding to MAb CR6261

To further determine if the autoreactivity observed for these bNAbs, using the protein microarray and SPR, impacts the binding between HA and the corresponding bNAb, we performed competition studies (using SPR) whereby the various self-proteins at 20 µg/mL were injected onto MAb-coated sensor chip, followed by injection of recombinant HA (10 μg/mL). The percentage of HA binding remaining in the presence of different host proteins were calculated with Bio-Rad ProteOn Manager software (version 3.0.1). As can be seen in Appendix A, binding of the bNAbs to a majority of self-antigens did not significantly reduce their subsequent binding to H1 (or H3) hemagglutinins. However, binding of EDC3 to MAb CR6261 specifically blocked subsequent interaction of the MAb to H1 HA (Figure 1C and Appendix A).

To identify the potential regions of homology in EDC3 with the known contact residues (DGW, QID, T, N, and I) for CR6261 on HA [5], homology modelling was performed. The potential sites of homology on the EDC3 sequence are highlighted in Appendix A. These sites come together to form the potential binding site when mapped on the structure of EDC3 (Figure 1D).

This finding suggests that at least one of the stem-targeting bNAbs (CR6261) that binds to the Gp 1 hemagglutinin stem, also binds to an autoantigen, EDC3, with high affinity, that mimics the antibody-binding paratope–epitope interaction in the HA and host protein, EDC3.

## 4. Discussion

Several stem-targeting bNAbs are in clinical development for treatment of influenza but are yet to show any clinical benefit [15] (Lim et al., presented at 28th ECCMID, Madrid, 21–24 April, 2018). MAb CR6261 is one of thirteen human MAbs that were isolated from a combinatorial-display libraries [12] that were constructed from human IgM^+^ memory B cells of individuals vaccinated with the seasonal influenza vaccine and were found to have broad neutralizing activity against multiple strains that belong to group 1 influenza, encompassing H1N1 seasonal influenza and H5N1 avian influenza. Such broad cross-reactivity provided support for expectations that next generation influenza vaccines may induce similarly broad cross-reactive polyclonal antibodies that potentially may provide protection against future drifted strains and more importantly, against some avian strains with pandemic potential [1,7]. Importantly, MAb CR6261 was shown to bind to the stem site using germline-encoded CDRs defined by IgV_H_1-69 via HCDR2 (containing F54 allele), which is directed to a hydrophobic pocket adjacent to the A helix with only small contribution from HCDR3 and none from the light chain [4]. This type of interaction is common to several stem-targeting bNAbs. Several studies demonstrated the CDRH2-centric contribution of the germline IgV_H_1-69, which provides natural affinity for the influenza virus group 1 epitope on the HA stalk [16,17]. It was therefore expected that immunogens that can bind to and activate B cells expressing this type of IgV_H_1-69 BCR could be expanded by suitable vaccination. However, to date, sustained expansion of broadly neutralizing anti HA stalk antibodies in humans following vaccination has not been achieved [2].

The hydrophobic CDRH2, which is encoded by the IgV_H_1-69, has been proposed to contribute to polyreactivity of several bNAbs [7]. Previous studies analyzed polyreactivity of MAbs in the absence or presence of BSA but did not evaluate either reactivity to human tissues or binding to human proteins in the presence of human serum, which is the natural milieu in vivo. Furthermore, the impact of binding to self-antigens on interaction of the bNAbs to its cognate influenza virus hemagglutinin was not addressed.

In the current study, we explored the potential autoreactivity of a panel of bNAbs targeting influenza virus hemagglutinin. Several bNAbs demonstrated binding to normal human tissues, but the reactivity was less prominent compared with HIV-1 MPER-specific MAb 4E10. The strongest reactivity was observed with brain pituitary gland tissue with three bNAbs CR6261, CR9114, and F2603. Moreover, using a ProtoArray microchip that contains > 9000 human proteins [13,18], we identified one autoantigen, EDC3, that bound with high affinity to bNAb CR6261 in the presence of human serum and that could compete with the binding of CR6261 to influenza virus hemagglutinin. This specific interaction likely reflects a true epitope mimicry between the host antigen and HA based on structural conservation in surface exposed regions (Figure 1D).

In a recent publication by Bajic et al., several other autoantigens were found to bind to CR6261 (UBE3A, NP-BSA, and Jo-1) [8]. One of these proteins, Jo-1, was also identified in our study, but it was bound by all the MAbs in our panel in the presence of Ig-depleted human serum, including Palivizumab (anti-RSV F site II), which was not shown to be autoreactive. However, several of proteins that were identified in the Bajic study, were not preferentially recognized by bNAbs in our study, which possibly may be due to the use of Ig depleted human sera as well as possibly high stringent cut-off used in this study. Moreover, Bajic et al. did not perform any protein-competition studies. Therefore, we reasoned that true autoreactivity of bNAbs should be demonstrated by competitive interaction of the human self-antigen with HA for binding to the bNAb. Indeed, when all the autoantigens identified in the ProtoArray analysis were used to block binding of bNAbs to HA in SPR, only EDC3 was shown to specifically block the binding of CR6261 to HA (Figure 1C and Appendix A). Such competition may reflect either common epitopes due to amino acid homology, structural mimicry, or steric hindrance. Homology analysis of EDC3 vs. HA identified contact residues of CR6261 that were mapped on the structure of EDC3 and may explain the observed high-affinity binding and competition with HA binding for MAb CR6261. It is unclear why the stem-targeting bNAb F10 which binds to a similar site on the HA stem as CR6261 did not bind to EDC3. This finding may reflect differences in affinity, fine epitope–paratope specificity, or interference by adjacent residues not shared between F10 and CR6261. Competition of bNAbs with the self-proteins does not necessarily correlate with the reactivity of the bNAbs to the proteins as measured by SPR, which suggests that some of the auto-reactivity may not be directly mediate via interactions with antibody paratope involved in binding to influenza HA.

The in vivo implication of autoreactivity associated with bNAbs that target the HA stalk remains to be determined. EDC3 is a component of a protein complex (P-body) responsible for mRNA decapping, which is essential for the controlled turnover of eukaryotic mRNA and regulation of gene expression. The crystal structure of human EDC3 has been resolved and shed light on its conformation and the sites required for mRNA binding and functions [10]. However, deciphering the potential impact of bNAb CR6261 binding of EDC3 will not be simple. EDC3 is part of a cytoplasmic complex that is unlikely to be exposed to antibodies outside the cell and may only be exposed in dying or lysed cells. Such constraints potentially present a hurdle to vaccine strategies that are designed to specifically expand these B cells and improve their affinity and durability [19], which needs to be further studied and/or taken into consideration.

In summary, some stem-specific bNAbs demonstrated autoreactivity to human tissues, and self-antigen EDC3 competed with HA binding to bNAb CR6261. These findings highlight the importance of carefully evaluating antibody-based therapeutics and next-generation influenza vaccines for autoreactivity with normal human tissues and identifying host self-antigens in conditions that mimic physiological environment in vivo.

## Figures and Tables

**Figure 1 viruses-12-01140-f001:**
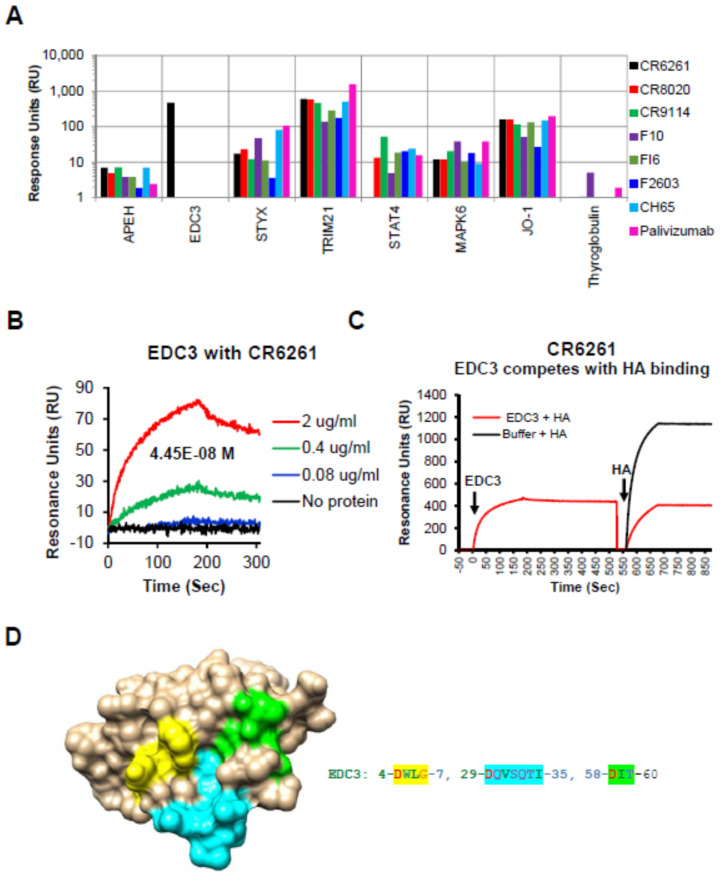
Reactivity of human proteins to bNAbs in Surface plasmon resonance (SPR): EDC3 blocks interaction between CR6261 and recombinant H1-HA. (**A**) SPR assay was performed with the MAbs captured on a protein-G sensor chip followed by addition of recombinant host protein (20 µg/mL) selected using ProtoArray. (**B**) Sensorgrams depicting the reactivity of serial dilutions of recombinant EDC3 in SPR to determine the binding affinity of recombinant EDC3 to the CR6261 by SPR. (**C**) Sensorgram to show competition between EDC3 and HA0 binding to CR6261. The binding of recombinant H1–HA0 (10 µg/mL) to the CR6261 bound EDC3 (10 µg/mL) was measured by SPR. Total H1–HA0 binding to CR6261 in absence of any EDC3 binding was defined as 100%. Percentage inhibition by the EDC3 was determined. (**D**) Structural depiction of potential binding site of CR6261 contact residues on EDC3 (PDB #2vc8). The EDC3 potential binding residue numbers and residues shaded in yellow, cyan, and green shown are depicted in similar colors on the EDC3 structure.

**Table 1 viruses-12-01140-t001:** Immunohistochemistry of MAbs on 30 Tissues microarray.

Mean (n = 3)	CR8020	CR6261	CR9114	F10	FI6	F2603	Palivizumab	CH65	4E10	Control
Adrenal	0.00	0.17	0.67	0.50	0.17	0.50	0.00	0.17	1.00	0.00
Bone Marrow	0.00	0.00	0.50	0.75	0.33	1.00	0.50	0.33	1.50	0.00
Breast	0.00	0.00	0.00	0.00	0.00	0.00	0.00	0.00	0.67	0.00
Brain, Cerebellum	0.00	0.00	0.00	0.00	0.00	0.00	0.00	0.00	0.00	0.00
Brain, Cerebrum	0.00	0.17	0.50	0.17	0.33	0.17	0.00	0.00	1.67	0.00
Brain, pituitary	0.00	1.33	1.67	0.00	0.17	1.50	0.00	0.00	2.83	0.00
Colon	0.00	0.00	0.17	0.00	0.00	0.17	0.00	0.00	0.83	0.00
Oesophagus	0.00	0.17	0.00	0.00	0.00	0.17	0.33	0.00	0.17	0.00
Heart	0.00	0.50	0.33	0.50	0.17	0.50	0.17	0.00	1.00	0.00
Kidney	0.17	0.33	0.00	0.33	0.00	0.17	0.00	0.00	1.17	0.00
Liver	0.00	0.33	0.33	0.17	0.17	0.33	0.17	0.00	2.33	0.00
Lung	0.00	0.00	0.00	0.00	0.17	0.33	0.17	0.00	0.17	0.00
Mesothelial Cell	0.00	0.17	0.00	0.00	0.00	0.00	0.00	0.17	0.33	0.00
Ovary	0.00	0.00	0.00	0.00	0.00	0.00	0.00	0.00	0.33	0.00
Pancreas	0.00	0.67	0.67	0.17	0.00	0.33	0.17	0.17	2.50	0.00
Peripheral Nerve	0.00	0.00	0.00	0.00	0.00	0.17	0.00	0.00	0.00	0.00
Placenta	0.00	0.50	0.50	0.50	0.50	0.50	0.67	0.50	0.67	0.00
Prostate	0.00	0.00	0.00	0.00	0.00	0.00	0.00	0.00	1.00	0.00
Salivary Gland	0.00	0.00	0.00	0.00	0.00	0.00	0.00	0.00	0.67	0.00
Skin	0.00	0.17	0.00	0.00	0.00	0.17	0.00	0.17	0.00	0.00
Small Intestine	0.00	0.17	0.00	0.00	0.00	0.00	0.00	0.00	1.33	0.00
Spleen	0.00	0.33	0.17	0.00	0.00	0.17	0.00	0.00	0.33	0.00
Skeletal muscle	0.00	0.17	0.00	0.00	0.00	0.00	0.00	0.00	0.00	0.00
Stomach	0.00	0.00	0.00	0.00	0.00	0.00	0.00	0.00	0.33	0.00
Testis	0.00	0.17	0.00	0.00	0.00	0.00	0.00	0.00	0.33	0.00
Thymus	0.00	0.00	0.00	0.00	0.00	0.00	0.00	0.00	0.17	0.00
Thyroid	0.00	0.00	0.17	0.00	0.00	0.00	0.17	0.00	0.00	0.00
Tonsil	0.00	0.00	0.00	0.00	0.00	0.00	0.00	0.00	0.00	0.00
Uterus	0.00	0.00	0.00	0.00	0.00	0.00	0.00	0.00	0.00	0.00
Uterus, Cervix	0.00	0.00	0.00	0.00	0.00	0.00	0.00	0.00	0.00	0.00

The color shading is heat map of MAb reactivity with human tissues.

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
