# Peer review of "Autoreactivity of Broadly Neutralizing Influenza Human Antibodies to Human Tissues and Human Proteins"

_viruses, 2020, doi:10.3390/v12101140_

Round 1
Reviewer 1 Report
Summary:
Measuring the potential for autoreactivity by reactivity to human autoantigens in vitro is an important metric for rank-ordering the potential clinical effectiveness of passively transferred antibodies. This can also inform on the feasibility of antibody vaccines concepts where the desired antibody response may also be limited by self-tolerance mechanisms. In the current study, the investigators apply these assays to examine 5-6 influenza bNAbs that target conserved features of the stem region on influenza hemagglutinin (HA) and have been considered inspirations for universal vaccine design. It has previously been established that HA stem bNAbs tend to be polyreactive and in some cases also bind human autoantigens in vitro. In this study, the investigators evaluate reactivity to even more comprehensive list of human autoantigens. Consistent with the previous literature, they observe reactivity of some (but not other) stem bNAbs to some of the autoantigens. A tight interaction (108M) between one of the bNAbs, CR6261, and the autoantigen EDC3 is also reported.
Major:
1)
As the authors correctly review, we already know that some monoclonal HA stem bNAbs are polyreactive and in some cases show affinity for autoantigens in vitro. It is true that this study does the most comprehensive screening of human autoantigens yet examined, but the picture that emerges is the same: some stem bNAbs show reactivity to some autoantigens and others do not (see Andrews et al. 2015, see below which the authors also cite). The conceptual advance is therefore limited. The power in this study would be to apply their enhanced autoantigen screening approach to evaluate more than 5-6 stem bNAbs (e.g. see Figure 7B in Andrews et al. 2015) so that general trends in autoreactive profiles could be more comprehensively catalogued and profiled. In this way, the field would be far better equipped to define whether autoreactivity is a systemic issue with stem-based bNAbs or not. But with this limited bNAb set, and ‘one-hit-wonders’ like CR6261, the data doesn’t answer the bigger questions. Otherwise, experimentally the study is well executed
Andrews, S. F.; Huang, Y.; Kaur, K.; Popova, L. I.; Ho, I. Y.; Pauli, N. T.; Henry Dunand, C. J.; Taylor, W. M.; Lim, S.; Huang, M.; Qu, X.; Lee, J. H.; Salgado-Ferrer, M.; Krammer, F.; Palese, P.; Wrammert, J.; Ahmed, R.; Wilson, P. C., Immune history profoundly affects broadly protective B cell responses to influenza. Sci Transl Med 2015, 7, (316), 316ra192.
2)
The authors make a big point about CR6261 affinity for EDC3, but this is not seen for other IGHV1-69 stem bNAbs that have an extremely similar paratope (such as F10); hence it is seems likely that this tight interaction is CR6261-specific with limited generality. The authors conclude in their abstract that “findings suggest that there may be constraints on the expansion and maturation of CR6261- like bNAbs in vivo”. F10 is CR6261-like but does not show this same autoreactivity profile.
3) The literature is being referenced for points that it does not show. One cannot do this.
- A) “One of the properties of the IGHV1- 69 constrained HIV-1-specific antibodies is their propensity to be polyreactive and/or autoreactive, leading to their early removal via clonal deletion and/or central tolerance [9, 10]. ”
This statement is very misleading and it is factually incorrect. It would indeed be a big deal if IGHV1-69 usage itself was experimentally shown to pre-dispose for autoreactivity.
But here is actually what reference 9 and 10 refer:
Reference 9 = Yang et al. 2013
-This study is looking at antibody responses to the hydrophobic MPER epitope on HIV Env. It is true that a human MPER bNAb, 4E10 uses IGHV1-69, but there are many others (2F2, 10E8 etc) that target the same region but don’t use IGHV1-69. The autoreactivity from MPER bNAbs comes from the hydrophobic contact that these antibodies must make with the membrane surface, but this is not dependent on IGHV1-69 usage. Yang et al. (2013) examined the immunological relationship between having the proposed autoantigen recognized by 2F2 and 4E10 MPER (present in most vertebrates) versus within opossums which lack a critical region of this autoantigen. The finding was that opossums are predisposed to elicit MPER-like responses and that this is may be related to the differences between the autoantigens. This study says nothing about usage of IGHV1-69, which is a human allele and is not present in opossums.
Reference 11 = Liu et al. 2015
-this study also does not examine or identify IGHV1-69 predisposition for autoreactivity. The major finding of this study is “four of nine bNAbs specific for the HIV-1 CD4 binding site (CD4bs) (VRC01, VRC02, CH106, and CH103) bind human ubiquitin ligase E3A (UBE3A), and UBE3A protein competitively inhibits gp120 binding to the VRC01 bNAb”. These antibodies are indeed constrained in VH, but not to IGHV1-69, rather these HIV bNAbs use IGHV1-2*02.
B) The important implication of our finding is the likelihood that certain naïve B cells expressing BCRs with strong autoreactivity will be subjected to clonal deletion at the pre-B/early B cells stage in the bone marrow or to subsequent peripheral tolerance. Influenza infection or vaccination of transgenic mice expressing humanized IGHV1-69 naïve B-cell repertoires elicited primary antibody response with stem-targeting antibodies, however, after subsequent exposures/vaccinations, most antibodies targeted other sites [8] These findings could have resulted from the immunodominance of the hemagglutinin globular head over the HA stalk.
Reference 8 = Sangesland et al. 2019
-this study demonstrated that IGHV1-69 naturally endows the B cell repertoire with BCRs that engage the bNAb target on the HA stem. This endowment was measured in the antigen naïve repertoire and the authors reported no evidence of clearance during B cell development. However, in IGHV1-69 mice, this endowed response was competed away by most HA immunogens, a property that could be reversed if the HA-head was removed by structure based design. I.e. with stalk-only HA immunogen, the investigators could now vaccine-expand serum bNAbs directed against the stalk. This response represented about 50% of the antigen specific response and was not cleared in the periphery. The authors should report as per the data described in the paper.
Author Response
Response to Reviewer:
Measuring the potential for autoreactivity by reactivity to human autoantigens in vitro is an important metric for rank-ordering the potential clinical effectiveness of passively transferred antibodies. This can also inform on the feasibility of antibody vaccines concepts where the desired antibody response may also be limited by self-tolerance mechanisms. In the current study, the investigators apply these assays to examine 5-6 influenza bNAbs that target conserved features of the stem region on influenza hemagglutinin (HA) and have been considered inspirations for universal vaccine design. It has previously been established that HA stem bNAbs tend to be polyreactive and in some cases also bind human autoantigens in vitro. In this study, the investigators evaluate reactivity to even more comprehensive list of human autoantigens. Consistent with the previous literature, they observe reactivity of some (but not other) stem bNAbs to some of the autoantigens. A tight interaction (108M) between one of the bNAbs, CR6261, and the autoantigen EDC3 is also reported.
Response: We appreciate reviewer’s comments to make our manuscript better
Major:
- As the authors correctly review, we already know that some monoclonal HA stem bNAbs are polyreactive and in some cases show affinity for autoantigens in vitro. It is true that this study does the most comprehensive screening of human autoantigens yet examined, but the picture that emerges is the same: some stem bNAbs show reactivity to some autoantigens and others do not (see Andrews et al. 2015, see below which the authors also cite). The conceptual advance is therefore limited. The power in this study would be to apply their enhanced autoantigen screening approach to evaluate more than 5-6 stem bNAbs (e.g. see Figure 7B in Andrews et al. 2015) so that general trends in autoreactive profiles could be more comprehensively catalogued and profiled. In this way, the field would be far better equipped to define whether autoreactivity is a systemic issue with stem-based bNAbs or not. But with this limited bNAb set, and ‘one-hit-wonders’ like CR6261, the data doesn’t answer the bigger questions. Otherwise, experimentally the study is well executed
Andrews, S. F.; Huang, Y.; Kaur, K.; Popova, L. I.; Ho, I. Y.; Pauli, N. T.; Henry Dunand, C. J.; Taylor, W. M.; Lim, S.; Huang, M.; Qu, X.; Lee, J. H.; Salgado-Ferrer, M.; Krammer, F.; Palese, P.; Wrammert, J.; Ahmed, R.; Wilson, P. C., Immune history profoundly affects broadly protective B cell responses to influenza. Sci Transl Med 2015, 7, (316), 316ra192.
Response: We performed an extensive analysis as possible based on the bNAbs available for this study. However, due to ongoing COVID-19 pandemic, we will not be able to perform additional experiments, since all our resources are now focused on COVID-19 research.
- The authors make a big point about CR6261 affinity for EDC3, but this is not seen for other IGHV1-69 stem bNAbs that have an extremely similar paratope (such as F10); hence it is seems likely that this tight interaction is CR6261-specific with limited generality. The authors conclude in their abstract that “findings suggest that there may be constraints on the expansion and maturation of CR6261- like bNAbs in vivo”. F10 is CR6261-like but does not show this same autoreactivity profile.
Response: Based on reviewer’s comment we have modified the abstract:
Abstract (Lines 38-40): These autoreactivity findings underscores the need for careful evaluation of such bNAbs for therapeutics and stem-based vaccines against influenza virus.
3) The literature is being referenced for points that it does not show. One cannot do this.
- A) “One of the properties of the IGHV1- 69 constrained HIV-1-specific antibodies is their propensity to be polyreactive and/or autoreactive, leading to their early removal via clonal deletion and/or central tolerance [9, 10]. ”
This statement is very misleading and it is factually incorrect. It would indeed be a big deal if IGHV1-69 usage itself was experimentally shown to pre-dispose for autoreactivity.
But here is actually what reference 9 and 10 refer:
Reference 9 = Yang et al. 2013
-This study is looking at antibody responses to the hydrophobic MPER epitope on HIV Env. It is true that a human MPER bNAb, 4E10 uses IGHV1-69, but there are many others (2F2, 10E8 etc) that target the same region but don’t use IGHV1-69. The autoreactivity from MPER bNAbs comes from the hydrophobic contact that these antibodies must make with the membrane surface, but this is not dependent on IGHV1-69 usage. Yang et al. (2013) examined the immunological relationship between having the proposed autoantigen recognized by 2F2 and 4E10 MPER (present in most vertebrates) versus within opossums which lack a critical region of this autoantigen. The finding was that opossums are predisposed to elicit MPER-like responses and that this is may be related to the differences between the autoantigens. This study says nothing about usage of IGHV1-69, which is a human allele and is not present in opossums.
Reference 11 = Liu et al. 2015
-this study also does not examine or identify IGHV1-69 predisposition for autoreactivity. The major finding of this study is “four of nine bNAbs specific for the HIV-1 CD4 binding site (CD4bs) (VRC01, VRC02, CH106, and CH103) bind human ubiquitin ligase E3A (UBE3A), and UBE3A protein competitively inhibits gp120 binding to the VRC01 bNAb”. These antibodies are indeed constrained in VH, but not to IGHV1-69, rather these HIV bNAbs use IGHV1-2*02.
Response: Based on reviewer’s comment we have modified the introduction:
Lines 63-66:
Interestingly, several HIV-1-specific bNAbs demonstrated propensity to be polyreactive and/or autoreactive{Haynes, 2005 #4641;Yang, 2013 #4630;Schroeder, 2006 #194;Liu, 2015 #4629;Bradley, 2016 #4643}.
- B) The important implication of our finding is the likelihood that certain naïve B cells expressing BCRs with strong autoreactivity will be subjected to clonal deletion at the pre-B/early B cells stage in the bone marrow or to subsequent peripheral tolerance. Influenza infection or vaccination of transgenic mice expressing humanized IGHV1-69 naïve B-cell repertoires elicited primary antibody response with stem-targeting antibodies, however, after subsequent exposures/vaccinations, most antibodies targeted other sites [8] These findings could have resulted from the immunodominance of the hemagglutinin globular head over the HA stalk.
Reference 8 = Sangesland et al. 2019
-this study demonstrated that IGHV1-69 naturally endows the B cell repertoire with BCRs that engage the bNAb target on the HA stem. This endowment was measured in the antigen naïve repertoire and the authors reported no evidence of clearance during B cell development. However, in IGHV1-69 mice, this endowed response was competed away by most HA immunogens, a property that could be reversed if the HA-head was removed by structure based design. I.e. with stalk-only HA immunogen, the investigators could now vaccine-expand serum bNAbs directed against the stalk. This response represented about 50% of the antigen specific response and was not cleared in the periphery. The authors should report as per the data described in the paper.
Response: To address reviewers concern, we have deleted the paragraph from the discussion section.

Reviewer 2 Report
This manuscript by Khurana et al. demonstrates that several broadly neutralizing influenza antibodies (bNAbs ) derived from VH1-69 sequences also react to self-proteins. These are potentially significant findings as several bNAbs are in clinical development and many universal vaccine platforms are being tested. The data presented in this manuscript highlight the importance of evaluating autoreactivity of these therapies. Two other groups also reported binding of bNAbs to self-proteins, and this manuscript extends these findings by analyzing reactivity to human tissue. In addition, they analyze binding of bNAbs to protein microarrays in the presence of Ig-depleted human serum, rather than BSA. Finally, they demonstrate competition of the CR6261 mAb with the self-protein, EDC3. These data are important because they independently demonstrate potential autoreactivity of bNAbs, and more rigorously test the binding of CR6261 to EDC3 and H1.
A few minor issues should be addressed to strengthen the conclusions:
- To better understand the range of scores assigned to the binding of the bNAbs to the human tissues, representative images of each score should be shown in supplemental Fig. 1.
- The proteins listed in Table S1 should indicate which antibody is binding to each protein.
- For the ProtoArray analysis, it would be helpful to see the binding pattern of each bNAb relative to the control 151k.
- The current study used the ProtoArray to test several antibodies that were previously tested by Bajic et al. on this same ProtoArray. However, several of proteins that were identified as binding the bNAbs in the Bajic study were not reported here. The authors should discuss whether this may be due to the use of different blocks or if different cut-offs were established.
- The authors should address why competition of the bNAbs with the self-proteins does not necessarily correlate with the reactivity of the bNAbs to the proteins as measured by SPR.
Author Response
Response to Reviewers’ comments:
Reviewer #1:
This manuscript by Khurana et al. demonstrates that several broadly neutralizing influenza antibodies (bNAbs ) derived from VH1-69 sequences also react to self-proteins. These are potentially significant findings as several bNAbs are in clinical development and many universal vaccine platforms are being tested. The data presented in this manuscript highlight the importance of evaluating autoreactivity of these therapies. Two other groups also reported binding of bNAbs to self-proteins, and this manuscript extends these findings by analyzing reactivity to human tissue. In addition, they analyze binding of bNAbs to protein microarrays in the presence of Ig-depleted human serum, rather than BSA. Finally, they demonstrate competition of the CR6261 mAb with the self-protein, EDC3. These data are important because they independently demonstrate potential autoreactivity of bNAbs, and more rigorously test the binding of CR6261 to EDC3 and H1.
Response: Reviewer’s kind words are highly appreciated.
A few minor issues should be addressed to strengthen the conclusions:
- To better understand the range of scores assigned to the binding of the bNAbs to the human tissues, representative images of each score should be shown in supplemental Fig. 1.
Response: Based on reviewer suggestion, we have added representative images for each bNAb with set of tissues on the array in revised supplemental figure 1.
- The proteins listed in Table S1 should indicate which antibody is binding to each protein.
Response: We have now included a representative binding of bNAbs with different proteins in the ProtoArray in new supplementary table 2.
- For the ProtoArray analysis, it would be helpful to see the binding pattern of each bNAb relative to the control 151k.
Response: Based on reviewer suggestion, we have provided the representative binding pattern of bNAbs with different proteins in the ProtoArray relative to control 151K in supplementary table 2.
- The current study used the ProtoArray to test several antibodies that were previously tested by Bajic et al. on this same ProtoArray. However, several of proteins that were identified as binding the bNAbs in the Bajic study were not reported here. The authors should discuss whether this may be due to the use of different blocks or if different cut-offs were established.
Response: This is possibly due to both the use of Ig depleted human sera and higher stringent cut-off in our study compared with Bajic et al. We have expanded the discussion.
Discussion (Lines 208-210 in tracked manuscript):
However, several of proteins that were identified in the Bajic study, were not preferentially recognized by bNAbs in our study, which possibly may be due to the use of Ig depleted human sera as well as high stringent cut-offs used in this study.
- The authors should address why competition of the bNAbs with the self-proteins does not necessarily correlate with the reactivity of the bNAbs to the proteins as measured by SPR.
Response: This is an important point and we have discussed it further.
Discussion (Lines 222-225 in tracked manuscript):
Competition of bNAbs with the self-proteins does not necessarily correlate with the reactivity of the bNAbs to the proteins as measured by SPR, which suggests that some of the auto-reactivity may not be directly mediate via interactions with antibody paratope involved in binding to influenza HA.
Round 2
Reviewer 1 Report
The Authors have addressed the concerns raised in my review.